# Egg Hatching and First Instar Falling Models of *Metcalfa pruinosa* (Hemiptera: Flatidae)

**DOI:** 10.3390/insects11060345

**Published:** 2020-06-03

**Authors:** Min-Jung Kim, Sunghoon Baek, Joon-Ho Lee

**Affiliations:** 1Entomology Program, Department of Agricultural Biotechnology, Seoul National University, Seoul 08826, Korea; 2017-24294@snu.ac.kr; 2Research Institute of Agriculture and Life Sciences, Seoul National University, Seoul 08826, Korea; bsh0627@hotmail.com

**Keywords:** *Metcalfa pruinosa*, overwintered eggs, egg hatching, nymph falling, degree-days model

## Abstract

Since the citrus flatid planthopper, *Metcalfa pruinosa* (Say), was introduced in Korea and many European countries, it has caused serious damage to various agricultural crops and landscape plants. *Metcalfa pruinosa* hibernates as eggs beneath the bark and in cracks of tree branches, and then substantial numbers of the first instar nymphs fall from the trees and move to other host plants. Knowing the timing of egg hatching and falling of the first instar nymphs would be key for controlling *M. pruinosa*. In this study, the hatching of overwintered *M. pruinosa* eggs and falling of the first instar nymphs from trees were monitored in several areas of Korea. These data were modeled with two starting points for degree-day accumulation, 1 January and 18 March, with a lower development threshold of 10.1 °C. The egg hatching and first instar falling models both used 1 January because the starting point performed better. The 50% appearance and falling times of the first instar nymphs were predicted to be 360.50 DD and 452.23 DD from 1 January, respectively, indicating that newly hatched nymphs stayed on the trees for about a week (i.e., 91.74 DD). Using these models, changes in the population density of the first instar nymphs of *M. pruinosa* on the trees were simulated, and the optimal control time range targeting the nymphs on the trees was deduced. The control time for nymphs on ground plants bordering the trees was suggested by the first instar falling model, along with observations of population density on the ground plants.

## 1. Introduction

The citrus flatid planthopper, *Metcalfa pruinosa* (Say) (Hemiptera: Flatidae), originating in eastern North America [1], has caused serious economic damage in many European countries and Korea [2,3]. In Europe, *M. pruinosa* was accidentally introduced first in Italy in 1979 [4], and then rapidly spread to the neighboring countries [2]. The rapid spread of *M. pruinosa* was also observed in Korea. *M. pruinosa* was discovered in almost all regions of Korea within 10 years from its first discovery in 2009 [3,5]. The high invasiveness of *M. pruinosa* appeared to be most strongly related to traffic factors [3,6,7]. In addition, its broad host range [8,9], high environmental adaptability [6], no natural enemy [10], and high susceptibility of host plants [11] in newly invaded areas might have contributed to its spread. Thus, the invasion and spread of *M. pruinosa* have persisted. Although *M. pruinosa* is not problematic in its native regions [12], it has caused serious economic impacts in invaded countries because of its mass occurrence [13]. In Italy, a yield loss of 30–40% of the soybean crop was reported [14]. In Korea, *M. pruinosa* has been a problem in various agricultural crops and fruit trees, such as ginseng, sesame, perilla, grape, peach, pear, persimmon, apple, and mulberry. [9,15]. *Metcalfa pruinosa* directly causes shoot stunting, plant vitality reduction, and plant wilting [6], and indirectly causes sooty molds on plants by excretion honeydews, which have also been considered a nuisance problem in urban areas [5].

*Metcalfa pruinosa* is univoltine, and hibernates as eggs beneath the bark and in the cracks of tree branches [6,16]. In spring, after the eggs are hatched, substantial numbers of first instar nymphs fall from the tree branches and move to herbaceous plants nearby [9,17]. The fallen nymphs persistently disperse to surrounding vegetation or crop fields, and this characteristic makes controlling *M. pruinosa* difficult. The adults return to the trees for oviposition soon after emergence [17], but are not proper targets for control because the sucking damage caused by the adults is less concerning than the nymphs [18] and the adults can easily escape chemical sprays by flight [19]. The effective target for the chemical control of *M. pruinosa* is probably the first instar nymphs clumping around the trees. In Korea, chemical spraying for the control of *M. pruinosa* has been implemented, targeting nymphs present in orchards, crop fields, and urban parks, and managers usually spray based on the observed nymph density. Wax filaments produced by growing nymphs could be a good indicator of the abundance of *M. pruinosa* nymphs [6], but they are hard to detect in newly hatched young nymphs due to their very small body size and low wax secretion. Thus, when the white wax of nymphs is detected, the nymphs could have already damaged the plants. Predictive studies for timing the appearance and movement of the first instar nymphs of *M. pruinosa* would be useful for deciding the optimal timing to control this pest in various landscapes because, despite being the main target of effective control, the first instar is most invisible.

Several studies have addressed the seasonal occurrence of *M. pruinosa* in temperate regions [8,13,17,19,20,21,22]. Lee et al. [22] reported the lower developmental threshold (LDT) of overwintered eggs to be 10.1 °C in laboratory conditions. They also proposed an egg hatching model of *M. pruinosa* with a starting degree-day accumulation date of April 1, for which no explanation was given but is needed. *Metcalfa pruinosa* nymphs moved from their overwintering trees to other host plants mainly from the end of May to the middle of June in a persimmon orchard [17]. However, no predictive model has been developed yet. Therefore, this study was conducted to (1) determine a proper starting degree-day accumulation point for models for the phenology of *M. pruinosa*, (2) develop an egg hatching model of *M. pruinosa* based on degree-days, (3) develop a falling model of the *M. pruinosa* first instar nymphs from the trees to the ground based on degree-days, and (4) discuss control options for *M. pruinosa* based on the simulation results using the developed models.

## 2. Materials and Methods

### 2.1. Data Collection for Model Development

The hatching of *M. pruinosa* overwintered eggs was monitored in Yeoncheon, Seoul, Suwon, and Yesan in Korea in 2018, and in Yeoncheon, Chuncheon, Seocheon, and Paju in Korea in 2019 to develop an egg hatching model (Table 1). The monitoring was carried out by observing the appearance of the first instar nymphs from samples of tree branches infested with *M. pruinosa* eggs. *M. pruinosa* adults prefer the trees with a rough surface of branches [23]; hence, this characteristic was considered for efficient sampling when the branch samples were collected. At each site, ten trees (e.g., *Robinia pseudoacacia*, *Zelkova serrata*, and *Cornus officinalis*) were selected, and five branches per tree were cut in late April. The excised branches were 10 cm in length from the base and > 1 cm in thickness because *M. pruinosa* adults prefer to lay eggs on thicker branches [23]. Then, five branches from the same tree were put into a nylon mesh pouch (30 × 20 cm, 104 × 94 mesh, BugDorm, Talchung, Taiwan). Thus, ten pouches of branches were prepared in each site. These pouches were put into a meshed fabric bag (90 × 90 cm, Eggthirty, Namyangju, Korea) to protect the nylon pouches. The bag was hung on a tree randomly selected at each site. The pouches were checked at one-week intervals for the appearance of *M. pruinosa* nymphs, except for the monitoring in Seoul in 2018, which was conducted every two days, from late April until no further egg hatching occurred. At each observation, the nymphs were counted and removed from the pouches.

The falling of the *M. pruinosa* first instar nymphs from the trees to the ground was monitored at the same sites as the egg hatching monitoring in 2018 and 2019 (Table 1). Clear, sticky panel traps (25 × 15 cm, Greenagrotech, Gyeongsan, Korea) were used for capturing the nymphs falling from the trees. The traps were installed 1 m above the ground around the trees in which the *M. pruinosa* eggs were found. At each site, three to six trees were randomly selected. Two rods (5.2 mm diameter × 160 cm height, Eden Plant Support, Gimhae, Korea) were embedded 40 cm vertically into the ground at a position 1 m away from each selected tree to fix the sticky traps. The distance between the two rods was the trap width, and the rods were fixed with connectors and metal sticks to maintain the width (Figure 1c). Then, the 50 cm tips of the rods were bent at 45° angles in the opposite direction of the trees, and a sticky trap was fixed on the tilted parts of two rods (Figure 1a,b). The reason for the tilted trap installation was to increase the capture capacity of the traps. The traps were changed on the same date as the egg hatching monitoring. The nymphs caught on the traps were counted under a microscope in the laboratory.

Air temperature data from the weather station closest to each monitoring site were obtained from the Korea Meteorological Administration website (http://web.kma.go.kr/). The temperature data were paired with field observation data of *M. pruinosa* egg hatching and nymph falling taken at each monitoring site. The daily maximum and minimum temperature data were used for degree-day calculations using the sine wave method.

### 2.2. Starting Point of Degree-Days Models

Two starting dates for the degree-day accumulation were examined to predict the phenology of first instar nymphs of *M. pruinosa*. One was 1 January, commonly used in degree-day models from a practical standpoint [24], and the other was a starting date empirically determined from our egg hatching data sets. The latter starting date was estimated as the point producing the lowest variation of accumulative degree-days to the first egg hatch for synchronizing the first appearance of nymphs by degree-days [25]. A total of 89 dates from February 1 to April 30 were potentially considered, and variations in the cumulative degree-days from the potential starting dates to the first egg hatch date of each monitoring site were compared. Egg hatching was not observed daily; therefore, the first egg hatching date of each monitoring site was estimated by extrapolating the first and second appearance of nymphs from the egg hatching data. The cumulative degree-days from each potential starting date were calculated using temperature data and an LDT of 10.1 °C for the *M. pruinosa* eggs. Among the potential 89 dates, a proper starting date was selected based on the coefficient of variation (CV) of the cumulative degree-days of eight data sets (two years × four sites). Finally, along with this empirically selected starting date, 1 January was examined as a starting point for the degree-day models.LDT

### 2.3. Estimation of Model Parameters

The eight data sets from the egg hatching and nymphal trap catch studies in the fields were converted to cumulative percent proportions of the first instar occurrence and falling, respectively. Each cumulative proportion (%) was fitted against the time (degree-days) accumulated from the two candidate starting points (i.e., 1 January and the selected date with the smallest CV) by the two-parameter Weibull function.
(1)$$F\left(x\right)=100\times \left[1-exp\left({\left(-x/\alpha \right)}^{\beta }\right)\right]$$
where F(x) is the cumulative proportion (%) of the appearance or falling of the first instars, for the respective models; time x is the degree-days, and α and β are the scale and shape parameters of the Weibull function, respectively. The parameters were estimated by using function ‘nls’ in R 3.5.1 [26]. Four models (egg hatching and first instar falling models with each of the two starting points) were finally constructed.

### 2.4. Validation and Accuracy of Models

Our independent field monitoring data were used to validate the models. The egg hatching data were collected in Yeoncheon (38°04′47.7″ N, 127°04′37.8″ E), Chuncheon (37°45′16.1″ N, 127°47′18.7″ E), Paju (37°59′05.1″ N, 126°58′59.5″ E), and Seocheon (36°11′03.8″ N, 126°46′20.1″ E) in 2019. The trap catch data were collected in Yeoncheon (38°04′47.7″ N, 127°04′37.8″ E) in 2017 and Seoul (37°27′27.4″ N, 126°56′54.4″ E) in 2019. All validation data from the fields were obtained by the same method described previously at one-week intervals. In addition, we used the 50% appearance times of the first instar nymphs at the three sites in Korea from the previous study [22].

To determine a starting date for the models that better described both the appearance and falling of the first instar nymphs, we grouped the models into two groups, one using 1 January and one with a potential date selected by the CV comparison. Then, the two groups were compared. We examined whether the predictive performance differed according to different starting points. For this, Julian dates corresponding to the cumulative proportion of validation data were calculated by each model for the two groups. Linear regression analysis was then performed between the predicted dates of the two groups, and estimates of the parameters were calculated with 95% confidence intervals (CI). The closer the slope and intercept of the regression line are to one and zero, respectively, the closer the predictive ability is between the two groups of models [27]. Then, the root means square error (RMSE) of each group was calculated to select the starting point with better performance.
(2)$$\mathrm{RMSE}=\sqrt{\;\frac{\sum_{i=1}^{n}{\left(D_{predicted}-D_{observed}\right)}^{2}}{n}}$$
where $D_{predicted}$ is the Julian date predicted by the models, $D_{observed}$ is the actual Julian date of the validation data, and n is the number of observations. The models producing smaller RMSE’s were chosen to forecast the egg hatching and nymphal falling of *M. pruinosa*. However, if no significant difference in accuracy was found between the two groups (less than one day), the models created by the degree-days accumulated from 1 January were selected for generality.

### 2.5. Change of Nymphal Density

After the *M. p**ruinosa* eggs are hatched, the first instar nymphs move to the ground from the trees. Thus, their relative population on the trees during the period of the first instar stage could be estimated using the cumulative proportion of egg hatching and the cumulative proportion of the first instar nymphs moving from the trees to the ground. The relative population size of the *M. pruinosa* first instar nymphs on the trees at *x* degree-days was simulated by the following model.
(3)$$F_{3}\left(x\right)=F_{1}\left(x\right)-pF_{2}\left(x\right)$$
where $F_{3}\left(x\right)$ is the relative proportion (%) of the first instar population on the tree over time, and $F_{1}\left(x\right)$ and $F_{2}\left(x\right)$ are the cumulative proportions estimated from the egg hatching model and the falling model of the first instar nymphs, respectively. The parameter p is the falling proportion of the first instar nymphs of the total population. The $F_{3}\left(x\right)$ was simulated with different values of p ranging from 0.1 to 0.9 because the proportion of falling nymphs is unknown. The time range of the relative maximum population density on the trees with different values of p was calculated through this model.

In 2019, the actual density of *M. pruinosa* nymphs on herbaceous plants around the trees was observed in Yeoncheon and Chuncheon, where the hatching and falling of the first instar nymphs were monitored (Table 1). Five plant leaves were randomly selected within a meter from a tree trunk, and the number of nymphs on the leaves was counted by the naked eye. The sampled herbaceous plants were *Hosta* spp. in Yeoncheon and *Parthenocissus tricuspidata* in Chuncheon. A total of 30 plant leaves around six trees were examined on each sampling occasion. Nymph density monitoring on the ground plants was conducted every week from early May before the first emergence of adult *M. pruinosa*. Finally, we verified if changes in the nymph population on the ground could be described well by the constructed models.

## 3. Results

### 3.1. Starting Date of the Degree-Days Models

The CVs of the degree-days accumulated from the potential starting date to the estimated first egg hatch date for eight occurrence data sets varied according to the starting dates (Figure 2). The CV of the cumulative degree-days increased as the starting date of the degree-day accumulation, and the first egg hatch date became closer. The CV was the smallest on 18 March. Therefore, this date and 1 January were applied for model development and were compared for model performance.

### 3.2. Egg Hatching and First Instar Falling Models

All models with a starting date of 1 January or 18 March provided a good fit for egg hatching and the first instar falling of *M. pruinosa* (Table 2 and Table 3). The Julian dates predicted by the models with 1 January or 18 March as start dates were almost identical. Neither the slope or intercept of the regression line was significantly different from one and zero, respectively (i.e., 1:1 line) (*p* > 0.05; Table 3). The models with 1 January as a start date were better because they were closer to the observed data (Table 3). Therefore, the models with a starting point of 1 January were selected for forecasting the appearance and falling time of the first instar nymphs of *M. pruinosa.*

There was a long-time gap between the appearance and falling of the first instar nymphs of *M. pruinosa,* and this time gap gradually increased with increasing daily degree-days (Figure 3 and Figure 4). The time gap between 50% egg hatching and 50% first instar falling was 91.74 DD, which was about a week in Korea (Table 4).

### 3.3. Change in Nymphal Density

The relative population size on the trees changed depending on *p*, the falling proportion of the first instar nymphs from the trees to the ground (Figure 5). The maximum population density on the trees was estimated at 474 DD when *p* = 0.1, and 423 DD when *p* = 0.9. Therefore, the highest density of the first instar nymphs of *M. pruinosa* on the trees would be from 423 DD to 474 DD. 

On the ground cover, the density of the nymphs reached a peak at ca. 530 DD, and then decreased (Figure 6). This peak time coincided with the time of 90% proportion described by the falling model (Table 4) and did not overlap with the time of the highest density of first instar nymphs on the trees predicted by our simulation model (Figure 5).

## 4. Discussion

The starting point for degree-day accumulation is a critical component of a degree-day model for predicting insect phenology [28]. Ideally, the starting point has to be established at the point where the development of the target stage of the insect begins [28]. However, in many cases, the actual time when the development of overwintered insects begins is very hard to know. Thus, many researchers have proposed different starting points, such as empirical fixed calendar dates (e.g., the first date of a month such as January) or bio-fixed dates (e.g., the first trap catch, the first occurrence, or crop planting date) for the reliable prediction of target insect phenology [28]. In this respect, we examined 18 March as a candidate point that showed the lowest variation in accumulative degree-days to the first hatch of *M. pruinosa*. However, the performance of models using 1 January appeared to be better than those using 18 March (Table 3), although both models well-explained the phenology of the first instar nymphs of *M. pruinosa*. Therefore, we selected models using 1 January for the general application to forecast the phenology of the first instar nymphs of *M. pruinosa*. 

Our models showed that newly hatched nymphs of *M. pruinosa* did not fall immediately. Rather, most of the newly hatched nymphs stayed on the trees for a certain period and then fell to the ground. This delayed falling pattern of *M. pruinosa* nymphs matched our field observed nymphal density on the ground cover (Figure 6). In the predicted 90% hatching time of the eggs, 427 DD, the nymphal density on ground plants was relatively low, indicating that many nymphs still remained on the trees at that time. Therefore, if control actions needed to be implemented, the time when the first instar nymphs of *M. pruinosa* are gathered on trees would be an effective timing to control them. Through a model of population changes induced by both the appearance and falling of nymphs, we estimated that the control time to target nymphs on the trees was from 423 DD to 474 DD. During this time range, young nymphs would be present in high densities on the trees, and 88.5–99.3% of the *M. pruinosa* eggs would hatch to first instar nymphs that are no longer protected by eggshells or the bark of trees against chemical sprays. Therefore, the chemical application during this time period would reduce the potential damage of *M. pruinosa*. This action would be applicable in various environments, such as orchards, unmanaged trees around crop fields, and roadside trees.

In a preliminary study, we found that the frequency of nymphs falling from trees was negatively correlated with the distance from the trees, i.e., within 10 m (unpublished data). This indicates that the chance of nymphs directly falling onto crops is very low in fields where they are located more than 10 m from trees with overwintering habitats of *M. pruinosa*, whereas crop damage by nymphs would be inevitable in crop fields closely bordering overwintering habitats of *M. pruinosa*. Ciampolini et al. [14] reported that an infestation of *M. pruinosa* nymphs begins from trees bordering the fields and expands toward the center in soybean fields. Likewise, in our field observation, once the nymphs landed on the ground, they dispersed to other host plants gradually over time. The nymphal density on the ground plants around the trees peaked near the predicted time of 90% completion of first instars falling from the trees (535 DD), and then drastically decreased, probably due to the dispersal of the nymphs. Thus, control actions in the crop fields adjacent to trees around 535 DD may be another option. This action includes the application of chemicals at the efficient timing and locations to control the fallen nymphs in crop fields before they spread over other crop plants. It is also applicable to ground plants in a variety of landscapes where *M. pruinosa* massively appeared. In crop fields sufficiently far from overwintering habitats where the nymphs cannot directly land, the immigration pattern of nymphs into the fields might vary depending on the distance from the source populations on the trees. If the fallen nymphs randomly immigrate into crop fields just after falling, the immigration pattern of the nymphs might be similar to their falling pattern, and the application of chemicals on the edge of fields closest to the trees could be effective at the time when falling of the first instar nymphs is nearly finished. In contrast, if fallen nymphs from the trees do not immediately move to crops due to their own dispersal ability or distance to the fields, the immigration time into the fields will be delayed. Therefore, an optimal control timing and range for *M. pruinosa* in field crops needs to be improved by coupling with observations of *M. pruinosa* in the fields.

## 5. Conclusions

In summary, we developed two degree-day models to forecast the egg hatch and falling of first instar nymphs of *M. pruinosa*, and by using these models, we presented a model to describe the population change of *M. pruinosa* nymphs on trees and ground over time. For the reliable prediction of phenological events related to *M. pruinosa*, 1 January was determined as a starting date of the degree-day models. Finally, we discussed two options to control the timing and target sites of *M. pruinosa*, using a simulation model and field observations on the ground cover. The results showed that between 423 DD and 474 DD on the trees and 535 DD on the ground, plants bordering the trees were good control timings for *M. pruinosa* nymphs. Our results would be applicable in most temperate regions where the seasonal occurrence of *M. pruinosa* appears to be similar [8,13,17,19,20,21,22], with a few adjustments after validation, if necessary.

## Figures and Tables

**Figure 1 insects-11-00345-f001:**
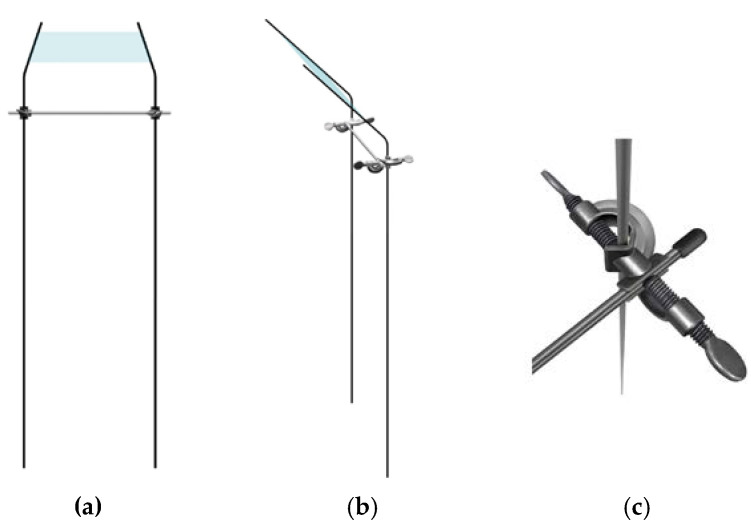
Installation of a clear sticky panel trap to capture the first instar nymphs falling from the trees (**a**) Front view; (**b**) Side view; (**c**) Connector to maintain the width of two rods as the trap width.

**Figure 2 insects-11-00345-f002:**
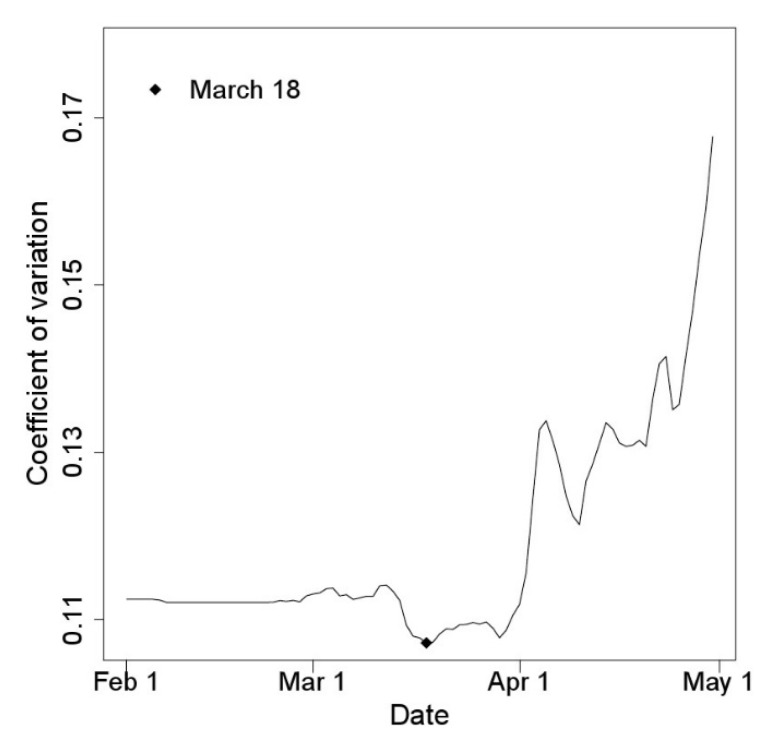
Coefficient of variation (CV) of the cumulative degree-days (DD) from the potential starting date to the first hatch of *M. pruinosa*; 18 March produced the lowest CV.

**Figure 3 insects-11-00345-f003:**
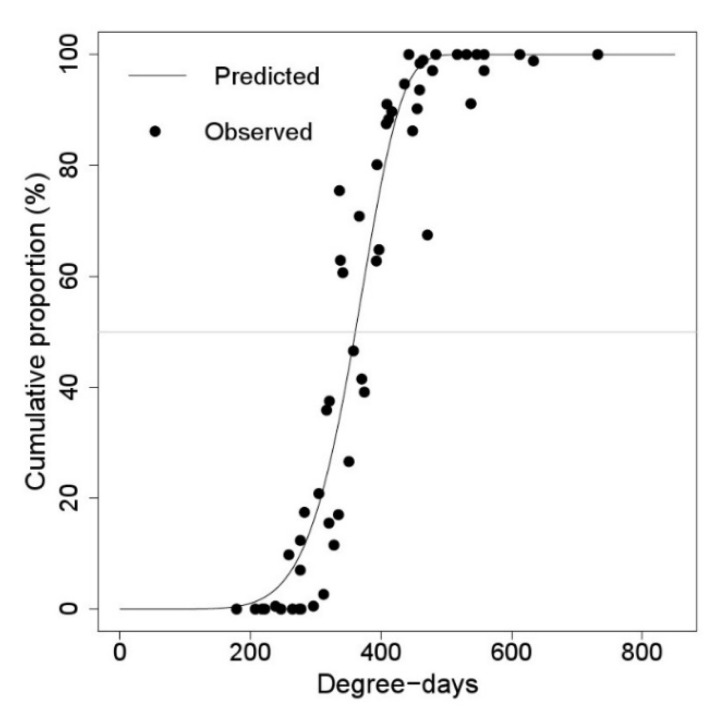
Distribution model of *M. pruinosa* egg hatching time, (DD) from 1 January.

**Figure 4 insects-11-00345-f004:**
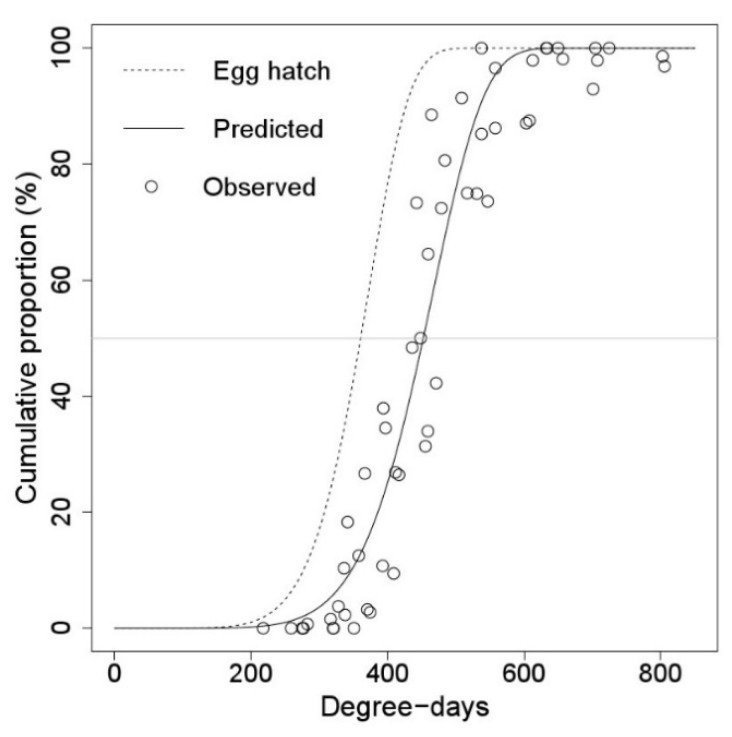
Distribution model of falling time of the first instar nymphs of *M. pruinosa* from the trees, (DD) from 1 January. The dashed line represents the egg hatching model of *M. pruinosa*.

**Figure 5 insects-11-00345-f005:**
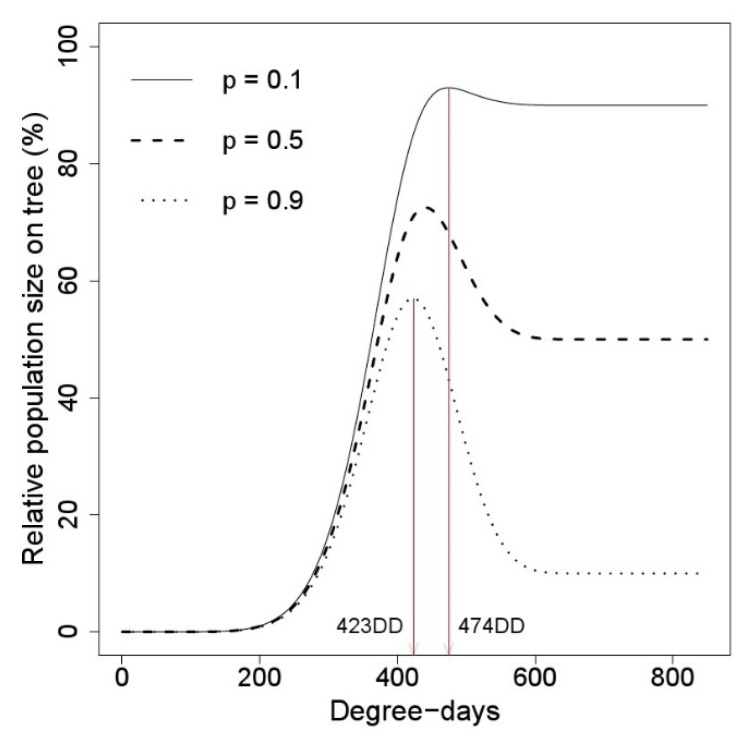
Simulation model for estimating the relative population density of *M. pruinosa* on the trees. The population on the trees was calculated with different moving *p* of the first instar nymph falling from the tree to the ground. The maximum density on the trees was estimated at 423 DD to 474 DD.

**Figure 6 insects-11-00345-f006:**
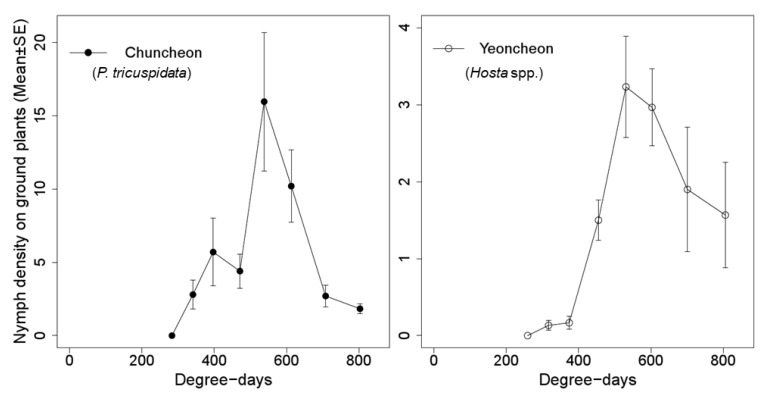
The mean density of *M. pruinosa* nymphs on 30 leaves of ground plants in Chuncheon and Yeoncheon.

**Table 1 insects-11-00345-t001:** Description of the study sites and sampling information for model development.

Site	Year	Latitude and Longitude (°, min, sec.)	Egg Hatching		First Instar Falling	
			No. Hatchings	First Appearance	No. Trap Catches	First Catch
Yeoncheon	2018	38°04′59.0″ N, 127°04′36.7″ E	194	5/16	85	5/29
Seoul	2018	37°27′27.4″ N, 126°56′54.4″ E	188	5/18	93	5/27
Suwon	2018	37°15′57.8″ N, 126°59′11.1″ E	24	5/21	15	5/28
Yesan	2018	36°44′11.9″ N, 126°49′17.9″ E	58	5/24	16	5/28
Yeoncheon	2019	38°04′59.0″ N, 127°04′36.7″ E	92	5/21	255	5/27
Chuncheon	2019	37°45′16.8″ N, 127°44′10.9″ E	338	5/20	142	5/20
Seocheon	2019	36°10′23.9″ N, 126°46′28.0″ E	156	5/22	53	5/22
Paju	2019	37°57′00.8″ N, 126°55′20.9″ E	171	5/21	29	5/27

**Table 2 insects-11-00345-t002:** Estimated parameter values for the distribution model of egg hatching time and first instar falling time of *M. pruinosa*.

Model	Starting Date ^1^	Parameter	
		α (Mean ± SEM)	β (Mean ± SEM)
Egg hatching	1 January	379.554 ± 4.4365 ^***^	7.115 ± 0.7276 ^***^
	18 March	362.466 ± 4.2109 ^***^	6.907 ± 0.6798 ^***^
First instar falling	1 January	476.119 ± 5.2429 ^***^	7.121 ± 0.6990 ^***^
	18 March	460.282 ± 5.3532 ^***^	6.819 ± 0.6776 ^***^

^1^ Starting date for degree-days (LDT = 10.1 °C) accumulation in each model. ^***^ All estimated parameter values significantly different from zero (*p* < 0.001).

**Table 3 insects-11-00345-t003:** Parameter values of the regression line between the Julian date predicted by the two groups of models with different starting points and the accuracy of each group.

Starting Date	N ^1^	Linear Regression				RMSE ^3^ (Days)
		Slope (95% CI ^2^)	Intercept (95% CI)	*p*-Value	r ^2^	
1 January	34	0.99 (0.98, 1.01)	1.14 (−1.72, 4.00)	<0.001	0.997	6.0
18 March						6.3

^1^ number of observations; ^2^ confidence intervals for the parameter of the regression line. ^3^ root means squared error of the Julian dates between the model-predicted and observed.

**Table 4 insects-11-00345-t004:** Predicted time (DD) of egg hatching and first instar falling of *M. pruinosa*.

Model	Cumulative Proportion (%)				
	10	30	50	70	90
Hatching	276.64	328.36	360.50	389.59	426.76
Falling	347.12	411.95	452.23	488.69	535.28
Difference	70.47	83.59	91.74	99.11	108.52

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
