# Peer review of "Egg Hatching and First Instar Falling Models of Metcalfa pruinosa (Hemiptera: Flatidae)"

_insects, 2020, doi:10.3390/insects11060345_

Round 1

Reviewer 1 Report

This paper presents models for describing the occurrence and disappearance of first instar nymphs of Metchlfa pruinose, a pest flatid insect, on the host plant. The models aims to control the first instar nymphs of the pest insect effectively. I think that the method is adequate, and this method can be used worldwide.

I have found some insufficient explanations in the text, but these points are all minor. If the authors revise these point, the paper is worth publishing.

Line 172-173. F1(x) and F2(x) should be “cumulative proportions” estimated from the egg hatching model and the falling model, respectively.

Line 173. p is a little difficult to imagine. If p <1, I think that some nymphs remain on the host tree, where they develop to adulthood. Is this speculation right?

Line 182. hosta should be Hosta. spp. should not be italic.

Line 214. I think that this time gap gradually increased “with increasing degree days”.

Line 278. The following sentence seems to be insufficient. “This action includes the efficient timing and locations…”

Consider the following “This action includes chemical application in the efficient timing and locations…”

Author Response

Dear Editor and reviewer,

We would like to thank the reviewers for valuable comments on our manuscript. We revised the manuscript according to reviewer’s comments. All changes in the revised version of manuscript used "Track changes" function in Microsoft Word.

General comment:

This paper presents models for describing the occurrence and disappearance of first instar nymphs of Metchlfa pruinosa, a pest flatid insect, on the host plant. The models aim to control the first instar nymphs of the pest insect effectively. I think that the method is adequate, and this method can be used worldwide.

I have found some insufficient explanations in the text, but these points are all minor. If the authors revise these point, the paper is worth publishing.

Our response: We appreciate your positive feedback. We corrected and made detailed explanations in the revised version. Followings are our responses to the specific comments.

Specific comment:

Line 172-173: F1(x) and F2(x) should be “cumulative proportions” estimated from the egg hatching model and the falling model, respectively.

Our response: We agree with your comment. We revised this paragraph reflecting your suggestion and made expression more clear.

Line 173: p is a little difficult to imagine. If p <1, I think that some nymphs remain on the host tree, where they develop to adulthood. Is this speculation right?

Our response: Since our simulation model accounts for occurrence and disappearance of only the first instar nymphs on the host trees, the parameter p is defined as the falling proportion of nymphs during developmental stage of the first instar. Some of nymphs fall from the trees after they become later  instar stage, and a few of them remain on the trees and can develop to adults on the trees. We revised this sentence more clearly.

Line 182: hosta should be Hosta. spp. should not be italic.

Our response: Thank you for pointing it out. We also corrected a legend in Figure 6.

Line 214: I think that this time gap gradually increased “with increasing degree days”.

Our response: We agree with your comment. The increase of the time gap may have arisen as the days warmer. We added the phrase “with increasing daily degree-days” in this sentence as you mentioned.

Line 278: The following sentence seems to be insufficient. “This action includes the efficient timing and locations…”

Consider the following “This action includes chemical application in the efficient timing and locations…”

Our response: The suggested correction was reflected to the revised version.

Reviewer 2 Report

This article is a sound presentation of modeling life history events of Metcalfa and should be published.  I detected no substantive issues with the manuscript, but have made several suggestions on the manuscript itself.  Small points include that the scientific names of plants should probably be included (the genus name Hosta should be capitalized), and the authors appeared to have redundantly presented gps locations where they could have cited Table 1.  

The English, and writing generally, is good, but could be improved by a careful read – mostly by deleted unneeded words or phrases. I did not attempt to edit the paper, but did provide occasional suggestions.

I think this paper can and should be published with few changes.

Author Response

Dear Editor and reviewer,

We would like to thank the reviewers for valuable comments on our manuscript. We revised the manuscript according to reviewer’s comments. All changes in the revised version of manuscript used "Track changes" function in Microsoft Word.

General comment:

This article is a sound presentation of modeling life history events of Metcalfa and should be published. I detected no substantive issues with the manuscript, but have made several suggestions on the manuscript itself. Small points include that the scientific names of plants should probably be included (the genus name Hosta should be capitalized), and the authors appeared to have redundantly presented gps locations where they could have cited Table 1.  

The English, and writing generally, is good, but could be improved by a careful read – mostly by deleted unneeded words or phrases. I did not attempt to edit the paper, but did provide occasional suggestions.

I think this paper can and should be published with few changes.

Our response: We appreciate your careful reading and positive feedback. We corrected our manuscript accordingly to improve readability. Our detailed responses are as follows.

Specific comment:

Line 13-16: suggest: Metcalfa pruinosa hibernates as eggs beneath bark and/or in cracks of tree branches, and then first instar nymphs falling from the trees and move to other host plants. Knowing the timing of egg hatching and falling of the first instar nymphs would be key for controlling M. pruinosa.

Our response: The suggested correction was made to the revised version.

Line 43: Editorial note - by convention you should spell out the genus when beginning a sentence (at least that is my understanding); but whether that is necessary is a decision for the editor.

Our response: Thank you for pointing it out. We corrected this and checked the whole manuscript.

Line 54: delete (this characteristic is in spring, not overwintering)

Our response: Done.

Line 79-80: How many of each; does the species matter? Add scientific names of plants (unless not required by journal). Starting when?

Our response: M. pruinosa lay eggs in most of tree species, but prefer trees with rough surfaces of branches than smooth ones. Thus we selected tree branches with rough surfaces for efficient sampling of M. puirnosa eggs. The tree species listed in this lines were mainly sampled species, which have rough surface and are very common in Korea. We added description of this background and sampling time in revised version. The common names of tree species were rewritten as scientific names.

Table 1: Delete bolding, subtending line

Our response: The table in original manuscript did not include bolding text and subtending, but it was automatically changed in uploading process. We checked uploaded file of revised version again.

Line 142-144: If these are presented in Table 1, are they needed here as well? suggest reference to that table instead.

Our response: Although the geographic names are the same, most of validation data were collected independently from the places where the data for model development shown in Table 1 were collected except for trap catches data in Seoul (same location in Table 1 but different year). So GPS locations are slightly different from Table 1 in this paragraph. Although it might be a little confusing, we still believe it is better to include in this paragraph. However, since the GPS location of the two sites where the actual density of nymphs was monitored were the same as Table 1, we replace them to reference of table for readability as you suggested.

Eq2: Spelling

Our response: Corrected.

Line 182-183: Capitalize, Scientific name

Our response: Corrected.

Figure 6: Capitalize 'h'

Our response: Corrected.

Line 270: spelling (spell-check change)

Our response: Corrected and checked the whole manuscript.
